# Piecewise Strong Convexity of Neural Networks

**Tristan Milne**
Department of Mathematics
University of Toronto
Toronto, Ontario, Canada
`tmilne@math.toronto.edu`

## Abstract

We study the loss surface of a feed-forward neural network with ReLU non-linearities, regularized with weight decay. We show that the regularized loss function is piecewise strongly convex on an important open set which contains, under some conditions, all of its global minimizers. This is used to prove that local minima of the regularized loss function in this set are isolated, and that every differentiable critical point in this set is a local minimum, partially addressing an open problem given at the Conference on Learning Theory (COLT) 2015; our result is also applied to linear neural networks to show that with weight decay regularization, there are no non-zero critical points in a norm ball obtaining training error below a given threshold. We also include an experimental section where we validate our theoretical work and show that the regularized loss function is almost always piecewise strongly convex when restricted to stochastic gradient descent trajectories for three standard image classification problems.

## 1  Introduction

Neural networks are an extremely popular tool with a variety of applications, from object ([SHK+14], [HZRS15]) and speech recognition [GMH13] to the automatic creation of realistic synthetic images [WLZ+17]. The optimization problem of finding the weights of a neural network such that the resulting function approximates a target function on training data is the focus of this paper.

Despite strong empirical success on this optimization problem, there is still much to know about the landscape of the loss function besides the fact that it is not globally convex. How many local minima does this function have? Are the local minima isolated from each other? What about the existence of local maxima or saddle points? We aim to answer several of these questions in this paper, at least for the loss function restricted to an important set of weights.

Important papers on the study of a neural network's loss function include [CHM+15] and [Kaw16]. In [CHM+15], the authors study the loss surface of a non-linear neural network. Under some assumptions, including independence of the network's input and the non-linearities, the loss function of the non-linear network can be reduced to a loss function for a linear network, which is then written as the Hamiltonian of a spin glass model. Recent ideas from spin glass theory are then used on the network's loss function, proving statements about the distribution of its critical values. This paper established an important first step in the direction of understanding the loss surface of neural networks, but leaves a gap between theory and practise due to its analysis of the expected value of the non-linear network over the behaviour of the ReLU non-linearities, which reduces the non-linear network to a linear one. Moreover, the authors assume that the components of the input data vector are independently distributed, which need not be true in important applications such as object recognition, where the pixel values of the input image are strongly correlated.

In [Kaw16], the results of [CHM$^+$15] are improved and several strong assumptions in that paper are either weakened or eliminated. This paper studies the loss surface of a linear neural network with a quadratic loss function, and establishes many strong conclusions, including the facts that every local minimum for such a network is a global minimum, there are no local maxima, and, under some assumptions on the rank of the product of weight matrices, the Hessian of the loss function has at least one negative eigenvalue at a saddle point, which has implications for the convergence of gradient descent via [LSJR16]. All this is proven under some mild assumptions on the distribution of the labelled data. These results, although comprehensive for linear networks, only hold for a network with ReLU activation functions if one first takes an expectation over the network's non-linearities as in [CHM$^+$15], and thus have limited applicability to non-linear networks.

Our contribution is to the understanding of the loss function for a neural network with ReLU non-linearities, quadratic error, and weight decay regularization. In contrast to [CHM$^+$15] and [Kaw16], we do not take an expectation over the network's non-linearities, and instead study the network in its fully non-linear form. Without making any assumptions on the labelled data distribution, we prove that the loss function has no local maxima in any open neighbourhood where the loss function is infinitely differentiable and non-constant. We also prove that there is a non-empty open set where

1. the regularized loss function is piecewise strongly convex, with domains of convexity determined by the ReLU non-linearity,

2. every differentiable critical point of the regularized loss function is a local minima, and,

3. local minima of the regularized loss function are isolated.

This open set contains the origin, all points in a norm ball where training error is below a given threshold, and under some conditions, all global minimizers of the regularized loss function. Our results also provide an explicit description of the set where piecewise strong convexity is guaranteed in terms of the size of the network's parameters and architecture, which allows for networks to be designed to maximize the size of this set. This set also implicitly depends on the training data. We emphasize the similarity of these results to those proved in [CHM$^+$15], which include the fact that there is a value of the Hamiltonian (i.e. the loss function of the network), below which critical points have a high probability of being low index. Our results find a threshold of the same loss function under which every differentiable critical point of the regularized function in a bounded set is a local minimum, and therefore has index $0$. Since we make no assumptions on the distribution of training data, we therefore hope that our results provide at least a partial answer to the open problem given in [CLA15].

We also include an experimental section where we validate our theoretical results on a toy problem and show that restricted to stochastic gradient descent trajectories, the loss function for commonly used neural networks on image classification problems is almost always piecewise strongly convex. This shows that even though the loss function may be non-convex outside the open set we have found, its gradient descent trajectories are similar to those of a strongly convex function, which hints at a general explanation for the success of first order optimization methods applied to neural networks.

## 1.1   Related Work

A related paper is [HM16], which shows that for linear residual networks, the only critical points in a norm ball at the origin are global minima, provided the training data is generated by a linear map with Gaussian noise. In contrast, we show that for a non-linear network with any training data, there is an open set containing the origin where every differentiable critical point is a local minimum.

Our result on the existence of regions on which the loss function is piecewise well behaved is similar to the results in [SS16], which shows that for a two layer network with a scalar output, weight space can be divided into convex regions, and on each of these the loss function has a basin-like structure which resembles a convex function in that every local minimum is a global minimum. Compared to [SS16], our conclusions apply to a smaller set in weight space, but are compatible with more network architectures (e.g. networks with more than two layers).

Another relevant paper is [SC16], which proves that for neural networks with a single hidden layer and leaky ReLUs, every differentiable local minimum of the training error is a global minimum under mild over-parametrization; they also have an extension of this result to deeper networks under stronger conditions. With small modifications to our proofs, our results can be applied to networks

with leaky ReLU's, and thus our paper and [SC16] provide complementary viewpoints; if every differentiable local minimum is a global minimum, then differentiable local minima must satisfy our error threshold as long as the network has enough capacity to obtain zero training error, and thus piecewise strong convexity is guaranteed around these points. Our results in combination with [SC16] therefore describe, in some cases, the regularized loss surface around differentiable local minima of the training error.

Convergence of gradient methods for deep neural networks has recently been shown in [AZLS18] and [OS19] with some restricted architectures. Since our results on the structure of the loss function are limited to a subset of weight space, they say little about the convergence of gradient descent with arbitrary initialization. Our empirical results in Section 4, however, show that the loss function is well behaved outside this subset, and indicate methods for generating new convergence results.

## 2    Summary of Results

Here we give informal statements of our main theorems and outline the general argument. In Section 3.1 we start by showing that a network of ReLU's is infinitely differentiable (i.e. smooth) almost everywhere in weight space. Using the properties of subharmonic functions we then prove that the corresponding loss function is either constant or not smooth in any open neighbourhood of a local maximum. In Section 3.2, we estimate the second derivatives of the loss function in terms of the loss function itself, giving the following theorem in Section 3.3.

**Theorem 1.** *Let $\ell$ be the loss function for a feed-forward neural network with ReLU non-linearities, a scalar output, a quadratic error function, and training data $\{a_i, f(a_i)\}_{i=1}^{N}$. Let*

$$\ell_\lambda(W) = \ell(W) + \frac{\lambda}{2}||W||^2,$$

*be the same loss function with weight decay regularization. Then there is an open set $U$, containing the origin, all points $W$ in a norm ball where training error is below a certain threshold, and, in certain cases, all global minimizers of $\ell_\lambda$, such that $\ell_\lambda$ is piecewise strongly convex on $U$.*

See Theorem 3 for a more precise statement of this result. In Section 3.4, we use piecewise strong convexity to prove the following theorem.

**Theorem 2.** *For the same network as in Theorem 1, every differentiable critical point of $\ell_\lambda$ in $U$ is a local minimum, and every local minimum of $\ell_\lambda$ in $U$ is an isolated local minimum.*

We conclude our theoretical section with an application of Theorem 2. We end with Section 4, where we empirically validate our theoretical results on a toy problem and demonstrate, for more complex problems, that the loss function is almost always piecewise strongly convex on stochastic gradient descent trajectories.

## 3    Theoretical Results

Consider a neural network with an $n_0$ dimensional input $a$. The neural network will have $H$ hidden layers, which means there are $H + 2$ layers in total, including the input layer (layer 0), and the output layer (layer $H + 1$). The width of the $i$th layer will be denoted by $n_i$, and we will assume $n_i > 1$ for all $i = 1, \ldots, H$. We will consider scalar neural networks for simplicity (i.e. $n_{H+1} = 1$), though these results can be easily extended to non-scalar networks. The weights connecting the $i$th layer to the $i + 1$st layer will be given by matrices $W_i \in \mathbb{R}^{n_i, n_{i+1}}$, with $w_{j,k}^i$ giving the weight of the connection between neuron $j$ in layer $i$ and neuron $k$ in layer $i + 1$. The neural network $y : \mathbb{R}^{n_0} \times \mathbb{R}^m \to \mathbb{R}$ is given by the function

$$y(a, W) = \sigma(W_H^T \sigma(W_{H-1}^T \sigma(\ldots \sigma(W_0^T a) \ldots))), \tag{1}$$

where $W = (W_0, \ldots, W_H)$ is the collection of all the network weights, and $\sigma(x_1, \ldots, x_n) = (\max(x_1, 0), \ldots, \max(x_n, 0))$ is the ReLU non-linearity. Let $f : \mathbb{R}^{n_0} \to \mathbb{R}$ be the target function, and let $\{a_i, f(a_i)\}_{i=1}^{N}$ be a set of labelled training data. The loss function is given by $\ell : \mathbb{R}^m \to \mathbb{R}$,

$$\ell(W) = \frac{1}{2N} \sum_{i=1}^{N} (f(a_i) - y(a_i, W))^2.$$

We will refer to $\ell(W)$ as the training error. The regularized loss function is given by $\ell_\lambda : \mathbb{R}^m \to \mathbb{R}$,

$$\ell_\lambda(W) = \ell(W) + \frac{\lambda}{2}||W||^2,$$

where $\lambda > 0$, and $||W||$ is the standard Euclidean norm of the weights. We will start by writing $y(a, W)$ as a matrix product. Define $S_i : \mathbb{R}^{n_0} \times \mathbb{R}^m \to \mathbb{R}^{n_i,n_i}$ by

$$S_i(a, W) = \mathrm{diag}(h_i(W_{i-1}^T \sigma(\ldots \sigma(W_0^T a) \ldots))), \qquad (2)$$

where $h_i : \mathbb{R}^{n_i} \to \mathbb{R}^{n_i}$ is given by

$$h_i(x_1, \ldots, x_{n_i}) = (1_{x_1 > 0}(x_1), \ldots, 1_{x_{n_i} > 0}(x_{n_i})),$$

where $1_{x_i > 0}$ is the indicator function of the positive real numbers, equal to 1 if the argument is positive, and zero otherwise. We will call the $S_i$ matrices the "switches" of the network. It is clear that

$$y(a, W) = S_{H+1}(a, W)W_H^T S_H(a, W)W_{H-1}^T S_{H-1}(a, W) \ldots S_1(a, W)W_0^T a.$$

## 3.1 Differentiability of the neural network

Here we state that the network is differentiable, at least on the majority of points in weight space.

**Lemma 1.** *For any $a \in \mathbb{R}^{n_0}$, the map $W \mapsto y(a, W)$ is smooth almost everywhere in $\mathbb{R}^m$.*

Although the proof of this lemma is deferred to the supplementary materials, the crux of the argument is simple; the network is piecewise analytic as a function of the weights, with domains of definition delineated by the zero sets of the inputs to the ReLU non-linearities. These inputs are themselves locally analytic functions, and the zero set of a non-zero real analytic function has Lebesgue measure zero; for a concise proof of this fact, see [Mit15].

Given that $y(a, W)$ is differentiable for almost all $W$, we compute some derivatives in the coordinate directions of weight space, and use the result to prove a lemma about the existence of local maxima.

**Lemma 2.** *If $W^*$ is a local maximum of $\ell$, then $\ell$ is not smooth on any open neighbourhood of $W^*$, unless $\ell$ is constant on that neighbourhood.*

This lemma is proved in the supplementary material, and involves showing that $\ell$ is a subharmonic function on any neighbourhood where it is smooth, and then using the maximum principle [McO03]. Note that the same proof applies to deep linear networks. These networks are everywhere differentiable, so we conclude that the loss functions of linear networks have no local maxima unless they are constant. This yields a simpler proof of Lemma 2.3 (iii) from [Kaw16].

## 3.2 Estimating Second Derivatives

This section will be devoted to estimating the second derivatives of $\ell$. This relates to the convexity of $\ell_\lambda$, since

$$\frac{d^2}{dt^2}|_{t=0} \ell_\lambda(W + tX) = \frac{d^2}{dt^2}|_{t=0} \ell(W + tX) + \lambda ||X||^2.$$

Hence, if there exists $\theta > 0$ such that

$$\frac{d^2}{dt^2}|_{t=0} \ell(W + tX) \geq -\theta ||X||^2,$$

then, provided $\lambda > \theta$, the loss function $\ell_\lambda$ will be at least locally convex. To estimate the second derivative of the loss function in an arbitrary direction, we first define the following norm, which measures the maximum operator norm of the weight matrices; it is the same norm used in [HM16].

**Definition 1.** *For a parameter $W \in \mathbb{R}^m$, define the norm*

$$||W||_* = \max_{0 \leq i \leq H} ||W_i||_2 \qquad (3)$$

*where $||\cdot||_2$ denotes the standard operator norm induced by the Euclidean norm.*

The proof of Lemma 1 shows that $\ell$ is everywhere equal to a loss function for a collection of linear neural networks which are obtained from $y(a_i, W)$ by holding the switches $\{S_j(a_i, W)\}_{i,j}$ constant. The next lemma estimates the second derivative of such a function in an arbitrary direction.

**Lemma 3.** *Suppose $\|a_i\|_2 \le r$ for all $1 \le i \le N$. Fix $W \in \mathbb{R}^m$, and set $\phi : \mathbb{R}^m \to \mathbb{R}$ as equal to $\ell$, but with switches $\{S_j(a_i, W)\}_{i,j}$ held constant as determined by $W$ and the dataset $\{a_i\}_{i=1}^N$. The second derivative of $\phi$ in direction $X \in \mathbb{R}^m$ satisfies*

$$\frac{d^2}{dt^2}\Big|_{t=0}\phi(W + tX) \ge -\sqrt{2}H(H+1)\|W\|_*^{H-1}\|X\|^2 r\phi(W)^{1/2}. \tag{4}$$

The proof of Lemma 3 is in the supplementary material, and uses standard tools.

## 3.3 Piecewise Convexity

Lemma 3 shows that the second derivative of $\ell$ in an arbitrary direction is bounded below by a term which depends on $\ell$. Thus, as the loss function gets small, the second derivative of the regularized loss function will be overwhelmed by the positive part contributed by the weight decay term. This observation leads to the following definition and theorem.

**Definition 2.** *For a fixed neural network architecture with $H$ hidden layers, and $r > 0$ satisfying*

$$\|a_i\|_2 \le r \quad 1 \le i \le N \tag{5}$$

*define the open set $U(\lambda, \theta)$, where $\lambda > \theta > 0$, by*

$$U(\lambda, \theta) = \{W \in \mathbb{R}^m \mid \ell(W)^{1/2}\|W\|_*^{H-1} < \frac{\lambda - \theta}{\sqrt{2}H(H+1)r}\}. \tag{6}$$

If we restrict to the norm ball $B(R) = \{W \in \mathbb{R}^m \mid \|W\|_* \le R\}$, we have

$$U(\lambda, \theta) \cap B(R) \supset \{W \in B(R) \mid \ell(W)^{1/2} < \frac{\lambda - \theta}{\sqrt{2}H(H+1)rR^{H-1}}\}. \tag{7}$$

Thus, $U(\lambda, \theta) \cap B(R)$ contains all points in $B(R)$ obtaining training error below the threshold given in (7); this is the inclusion referred to in Theorem 1. The set $U(\lambda, \theta)$ is non-empty provided $H > 1$, as it contains an open neighbourhood of $0 \in \mathbb{R}^m$. Further, $U(\lambda, \theta)$ contains all points $W$ obtaining zero training error, if these points exist; [ZBH+16] gives some sufficient conditions on the network architecture for obtaining zero training error. Lemma 4 gives conditions under which $U(\lambda, \theta)$ contains all global minimizers of $\ell_\lambda$.

**Lemma 4.** *Let $\epsilon = \inf_{W \in \mathbb{R}^m} \ell_\lambda(W)$. If*

$$\epsilon < \frac{\lambda^{1+1/H}}{2\left(H(H+1)r\right)^{2/H}}, \tag{8}$$

*then there exists $\theta$ such that $U(\lambda, \theta)$ contains all global minimizers of $\ell_\lambda$.*

Lemma 4 is proved in the supplementary materials. It follows since the term $\ell(W)^{1/2}\|W\|_*^{H-1}$ is roughly bounded by $\ell_\lambda(W)$. Observe that by evaluating $\ell_\lambda(W)$ at $W = 0$, we get the inequality

$$\epsilon \le \ell(0) = \frac{1}{2N}\sum_{i=1}^N f(a_i)^2. \tag{9}$$

Thus, as long as $\lambda$ is large enough to satisfy

$$\frac{1}{2N}\sum_{i=1}^N f(a_i)^2 < \frac{\lambda^{1+1/H}}{2\left(H(H+1)r\right)^{2/H}}, \tag{10}$$

Lemma 4 shows that there exists $\theta$ such that $U(\lambda, \theta)$ contains all global minimizers of $\ell_\lambda$. Thus, for any problem, we have found a range of $\lambda$ values where our set $U(\lambda, \theta)$ is guaranteed to contain all global minimizers of the problem.

Theorem 3 shows that we can characterise $\ell_\lambda$ on $U(\lambda, \theta)$.

**Theorem 3** (Piecewise Strong Convexity). *With $U(\lambda, \theta)$ defined as above, there exist closed sets $B_i \subset \mathbb{R}^m$ for $i = 1, \ldots, L$ such that*

$$U(\lambda, \theta) = \bigcup_{i=1}^{L} B_i \cap U(\lambda, \theta), \tag{11}$$

*and smooth functions $\phi_i : V_i \subset \mathbb{R}^m \to \mathbb{R}$, $V_i$ open, satisfying*

$$\mathbf{H}(\phi_i)(W) \geq \theta I_m, \forall W \in V_i \tag{12}$$

*where $\mathbf{H}(\phi_i)$ is the Hessian matrix, $B_i \cap U(\lambda, \theta) \subset V_i$ for all $i$, and*

$$\ell_\lambda|_{B_i \cap U(\lambda, \theta)} = \phi_i|_{B_i \cap U(\lambda, \theta)}. \tag{13}$$

The proof of Theorem 3 is in the supplementary materials. The sets $B_i$ are obtained from enumerating all possible values of the switches $\{S_j(a_k, W)\}_{j,k}$ as we vary $W$, and taking $B_i$ as the topological closure of all weights $W$ giving the $i$th value of the switches. The functions $\phi_i$ are obtained from $\ell_\lambda$ by fixing the switches according to the definition of $B_i$.

Note that this theorem implies Theorem 1, with $U$ given by $U(\lambda, \theta)$. This shows that when the training error is small enough in a bounded region, as prescribed by $U(\lambda, \theta)$ though (7), the function $\ell_\lambda$ is locally strongly convex. Note also that our estimates depend implicitly on the training data $\{a_i, f(a_i)\}_{i=1}^{N}$, and the widths of the network's layers, since these quantities affect the size of the set $U(\lambda, \theta)$.

### 3.4  Isolated Local Minima

The next two lemmas are an application of Theorem 3, and prove Theorem 2.

**Lemma 5.** *Every differentiable critical point of $\ell_\lambda$ in $U(\lambda, \theta)$ is an isolated local minimum.*

**Lemma 6.** *Every local minimum of $\ell_\lambda$ in $U(\lambda, \theta)$ is an isolated local minimum.*

The proofs of these lemmas are given in the supplementary materials. Note the subtle difference between the two; Lemma 5 only applies to critical points where $\ell_\lambda$ is differentiable, while Lemma 6 applies to non-differentiable local minima. Lemma 5 can be applied to prove that non-zero local minima obtaining training error below our threshold do not exist when the network is linear.

**Lemma 7.** *If $\ell_\lambda$ is the regularized loss function for a linear neural network with $H \geq 1$ and $n_i > 1$ for all $i = 1, \ldots, H$, then $\ell_\lambda$ has no non-zero critical points on the set $U(\lambda)$ given by*

$$U(\lambda) = \bigcup_{\theta > 0} U(\lambda, \theta) = \{W \in \mathbb{R}^m \mid \ell(W)^{1/2} \|W\|_*^{H-1} < \frac{\lambda}{\sqrt{2}H(H+1)r}\}. \tag{14}$$

This lemma is proved in the supplementary materials, and involves the use of rotation matrices to demonstrate that any non-zero local minimum in $U(\lambda)$ cannot be an isolated local minimum.

## 4  Experiments

We begin our experimental analysis with a regression experiment to validate our theoretical results; namely that the region $U(\lambda, \theta)$ is accessed over the course of training. We used a target function $f(x) = x/4$, with 100 data points sampled uniformly in $[-1, 1]$, and a neural network with $H = 1, n_1 = 2$, no biases, no ReLU on the output, weight decay parameter $\lambda = 0.4$, and learning rate $\eta = 0.1$. In this experiment, we measure the change in the regularized loss $\ell_\lambda$ from when gradient descent first enters $U(\lambda)$ to convergence, as a fraction of the total change in $\ell_\lambda$ from initialization to convergence; this quantity measures the portion of the gradient descent trajectory contained in $U(\lambda)$. We ran 1000 independent trials in PyTorch, distinguished by their random initializations, and found that the bound given in (6) is satisfied for some $\theta$ for $87.2\% \pm 22.3\%$ (mean $\pm$ standard deviation) of the loss change over training. A histogram of the loss change fractions over these trials is given in Figure 1 as well as a plot of the training error $\ell$ for a single initialization, selected to show the

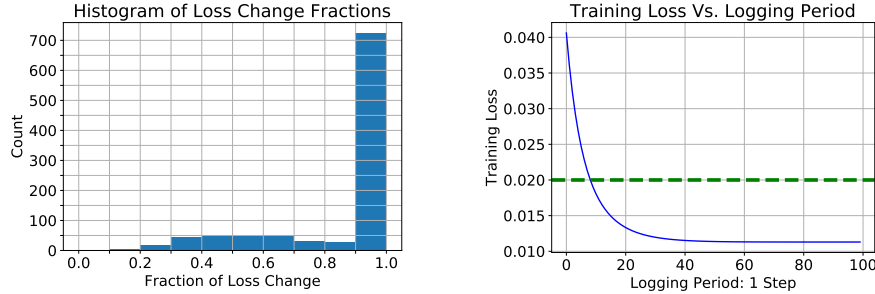

Figure 1: **Left**: Histograms of loss change fractions for $f(x) = x/4$ over 1000 trials. **Right**: A plot of the training loss $\ell$ versus time for a single training run on the same toy example. When the loss is below the horizontal dashed line, the current parameters $W$ are in $U(\lambda)$

trajectory entering $U(\lambda)$ over the course of training. The histogram in 1 shows that, for this simple experiment, most gradient descent trajectories inhabit $U(\lambda)$ for almost the entire training run.

For more complicated architectures, we found that the bound in (6) is difficult to satisfy while still obtaining good test performance. The goal of the rest of this section is therefore to empirically determine when the loss function for standard neural network architectures is piecewise convex, which may occur before the bound in (6) is satisfied. Because the Hessian of the loss function $\ell_\lambda$ is difficult to compute for neural networks with many parameters, we turn our focus to $\ell_\lambda$ restricted to gradient descent trajectories, which is similar to the analysis in [SMG13]. To that end, let $W : \mathbb{R}^+ \to \mathbb{R}^m$ be a solution to the ODE

$$\dot{W}(t) = -\nabla \ell_\lambda(W(t)), \quad W(0) = W_0, \tag{15}$$

and let $\gamma(t) = \ell_\lambda(W(t))$ be the restriction of $\ell_\lambda$ to this curve. We have

$$\ddot{\gamma}(t) = 2\nabla \ell_\lambda(W(t))^T \mathbf{H}(\ell_\lambda(W(t)))\nabla \ell_\lambda(W(t)). \tag{16}$$

If this is always positive, then $\ell_\lambda$ is piecewise convex on gradient descent trajectories. We are especially interested in the right hand side of (16) normalized by $\|\nabla \ell_\lambda\|^2$, due to Lemma 8.

**Lemma 8.** *Suppose that $\gamma(t)$ is $C^2$ over $[0, t^*] \subset \mathbb{R}$, and that*

$$\frac{2\nabla \ell_\lambda(W(t))^T \mathbf{H}(\ell_\lambda(W(t)))\nabla \ell_\lambda(W(t))}{\|\nabla \ell_\lambda(W(t))\|^2} \geq C. \tag{17}$$

*Then we have the convergence estimate*

$$\|\nabla \ell_\lambda(W(t))\|^2 \leq \|\nabla \ell_\lambda(W(0))\|^2 e^{-Ct} \quad \forall t \in [0, t^*]. \tag{18}$$

This lemma is proved in the supplementary material, and makes use of Grönwall's inequality. We may compute the second derivative of $\gamma(t)$ while avoiding computing $\mathbf{H}(\ell_\lambda)$ by observing that

$$2\nabla \ell_\lambda^T \mathbf{H}(\ell_\lambda)(W)\nabla \ell_\lambda = \nabla \ell_\lambda^T \nabla \|\nabla \ell_\lambda\|^2. \tag{19}$$

and therefore we may compute (16) by two backpropagations.

One notable difference between our theoretical work and the following experiments is that we focus on image classification tasks below, and hence the loss function is given by a composition of a softmax and cross entropy, as opposed to a square Euclidean distance.

For the MNIST, CIFAR10, and CIFAR100 datasets, we produce two plots generated by training neural networks using stochastic gradient descent (SGD). The first is of $\ell_\lambda$ and normalized second derivative, as given by the left hand side of (17), versus training time, represented using logging periods, calculated over a single SGD trajectory. The second is a histogram which runs the same test over several trials, distinguished by their random initializations, and records the percent change in $\gamma(t)$ between when it first became piecewise convex to convergence. In other words, let $t_0$ be the first

time after which $\ddot{\gamma}(t)$ is always positive; in our experiments, such a $t_0$ always exists. Let $t_1$ and $t = 0$ be the terminal and initial time, respectively. To compute the histograms in the left column of Figure 2, we compute the loss change fraction

$$\frac{\gamma(t_0) - \gamma(t_1)}{\gamma(0) - \gamma(t_1)} \tag{20}$$

over $n$ separately initialized training trials on a given dataset. We are interested in (20) as it measures how much of the SGD trajectory is spent in a piecewise convex regime.

Our experimental set-up for each data set is summarized in Table 1, and all experiments were implemented in PyTorch, with a mini-batch size of 128. Results are summarized in Table 2 in the form mean $\pm$ standard deviation, calculated across all trials for each data set. The column "Norm. 2nd. Deriv. (%10)" in Table 2 is the 10th percentile of the normalized second derivatives in the piecewise convex regime, as calculated within each trial; we present this instead of the minimum normalized second derivative as the minimum is quite noisy, and as such may not reflect the behaviour of the second derivative in bulk.

Table 1: Experimental Set-up

| Dataset | Model | Epochs | Batch Norm. | $\lambda$ | Trials | Learn. Rate |
|---------|-------|--------|-------------|-----------|--------|-------------|
| MNIST | LeNet-5 | 2 | No | $5 * 10^{-4}$ | 100 | 0.05 |
| CIFAR10 | ResNet22 | 65 | Yes | $1 * 10^{-6}$ | 20 | 0.1/0.02 |
| CIFAR100 | ResNet22 | 65 | Yes | $1 * 10^{-7}$ | 20 | 0.1/0.02 |

## 4.1 MNIST Experiments

In our first experiment we used the LeNet-5 architecture [LBB+98] on MNIST. The histogram generated for MNIST in Figure 2 shows that the loss function is piecewise convex over most of the SGD trajectory, independent of initialization. The upper right plot of Figure 2 confirms this; the normalized second derivative, in red, is negative for a very small amount of time, and then is consistently larger than 10, indicating that the loss function is piecewise strongly convex on most of this SGD trajectory. This is similar to our theoretical results as piecewise strong convexity only seems to occur after the loss is below a certain threshold.

## 4.2 CIFAR10 and CIFAR100 Experiments

This group of experiments deals with image classification on the CIFAR10 and CIFAR100 datasets. We trained the ResNet22 architecture with PyTorch, and produced the same plots as with MNIST. This version of ResNet22 for CIFAR10/CIFAR100 was taken from the code accompanying [FOA18], and has a learning rate of 0.1, decreasing to 0.02 after 60 epochs.

In the middle left histogram in Figure 2, we see that over 20 trials on CIFAR10, every SGD trajectory was in the piecewise convex regime for its entire course. This is again reflected in the middle right plot of Figure 2, where we also see the normalized second derivative being consistently larger than 10. The test set accuracies are given in Table 2 for all three data sets, and are non-optimal; this is intentional, as we wanted to study the loss surface without optimization of hyperparameters. Note that Table 2 shows that large values of the normalized second derivative are consistently observed across all initializations, and for every dataset.

Table 2: Experimental Results

| Dataset | Test Acc. (%) | Norm. 2nd. Deriv. (%10) | Loss Frac. |
|---------|---------------|-------------------------|------------|
| MNIST | $97.32 \pm 1.00$ | $21.19 \pm 2.12$ | $0.80 \pm 0.09$ |
| CIFAR10 | $84.93 \pm 0.35$ | $8.82 \pm 2.80$ | $1.00 \pm 0.00$ |
| CIFAR100 | $54.01 \pm 0.53$ | $11.80 \pm 0.27$ | $1.00 \pm 0.02$ |

To see if this behaviour is consistent with more challenging classification tasks, we tested the same network on CIFAR100; the results are summarized in the bottom row of Figure 2, and are consistent with CIFAR10, with only 1 trajectory out of 20 failing to be entirely piecewise convex. We conjecture

that the extra convexity observed in CIFAR10/100 compared to MNIST is due to the presence of batch normalization, which has been shown to improve the optimization landscape [STIM18].

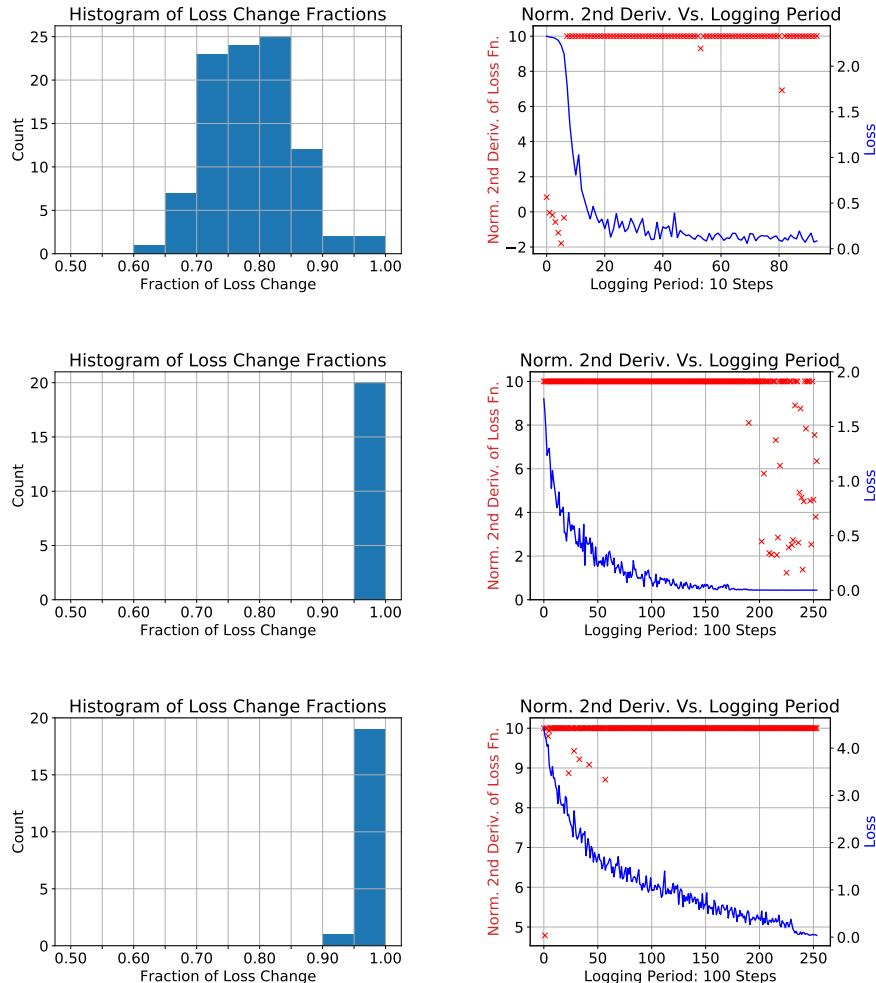

Figure 2: **Left**: Histograms of loss change fractions for three image classification problems over several trials. **Right**: A plot of the loss function and normalized second derivative versus time for a single training run on the same datasets. The normalized second derivative is clipped at 10 for legibility. **Top**: MNIST, **Middle**: CIFAR10, **Bottom**: CIFAR100. Best viewed in color.

# 5   Conclusion

We have established a number of facts about the critical points of the regularized loss function for a neural network with ReLU activation functions. In particular, there are, in a certain sense, no differentiable local maxima, and, on an important set in weight space, the loss function is piecewise strongly convex, with every differentiable critical point an isolated local minimum. We then applied this result to prove that non-zero local minima of a regularized loss function for linear networks obtaining training error below a certain threshold do not exist. Finally, we established the relevance of our theory to a toy problem through experiments, and demonstrated empirically that the loss function for standard neural networks is piecewise strongly convex along most SGD trajectories. Future directions of research include investigating if the re-parametrization offered by Residual Networks [HZRS15] can improve our analysis, as was observed in [HM16], as well as determining where $\ell_\lambda$ satisfies (17), which would allow for stronger theorems on the convergence of SGD.

# 6 Acknowledgements

We would like to thank the anonymous referees for their thoughtful reviews. Thanks also to Professor Adam Stinchcombe for the use of his GPUs. This work was partially supported by an NSERC PGS - D and by the Mackenzie King Open Scholarship.

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
