[Supplementary Material · Piecewise Strong Convexity - SMFINAL.pdf]

# Piecewise Strong Convexity of Neural Networks Supplementary Materials

**Tristan Milne**
Department of Mathematics
University of Toronto
Toronto, Ontario, Canada
`tmilne@math.toronto.edu`

## 1 Supplementary Materials

Here we will give detailed proofs for the results given in the main paper. Any equation reference with the prefix MP- indicates a reference to an equation number in the main paper. We start with the claim that the network in consideration is differentiable almost everywhere in weight space.

**Proof of Lemma 1**. Note that the claim is trivial if $a = 0$, so we proceed assuming $a \neq 0$. First, define $S_i' : \mathbb{R}^{n_0} \times \mathbb{R}^m \to \mathbb{R}^{n_i}$ by

$$S_i'(a, W) = h_i'(W_{i-1}^T \sigma(\dots \sigma(W_0^T a) \dots)),$$

where $h_i' : \mathbb{R}^{n_i} \to \mathbb{R}^{n_i}$ is given by

$$h_i'(x_1, \dots, x_{n_i}) = (\text{sign}(x_1), \dots, \text{sign}(x_{n_i})),$$

where

$$\text{sign}(x) = \begin{cases} 1 & x > 0, \\ 0 & x = 0, \\ -1 & x < 0. \end{cases}$$

First, we claim that $W \mapsto y(a, W)$ is smooth at $W^*$ if all the elements of $S_i'(a, W^*)$ are non-zero for all $i$. Indeed, if this is the case, then, for $W$ in an open neighbourhood of $W^*$,

$$y(a, W) = W_H^T S_H(a, W^*) W_{H-1}^T S_{H-1}(a, W^*) \dots S_1(a, W^*) W_0^T a, \tag{1}$$

which is a polynomial function of $W$, and so is smooth. So, we may proceed assuming that at $W^*$, at least one element of $S_i'(a, W^*)$ is zero for some $i$. Write $S_{i,j}'(a, W^*)$ as this element, so that the index $i$ refers to the layer, and $j$ refers to the neuron in that layer. We may proceed without loss of generality assuming that $S_{k,l}'(a, W^*) \neq 0$ for any $k < i$ and all $l$, since otherwise we could relabel $S_{i,j}'(a, W^*)$ as $S_{k,l}'(a, W^*)$ where $k$ is minimal. As such, $W^*$ is in the set

$$A = \{W \in \mathbb{R}^m \mid \exists i, j, \text{ such that } S_{i,j}'(a, W) = 0, \text{ and } S_{k,l}'(a, W) \neq 0$$
$$\forall k < i, l \in \{1, \dots, n_k\}\}.$$

Let us partition $A$ into two subsets, $B$ and $C$, where

$$B = \{W \in A \mid S_{i-1,j,j}(a, W) = 0 \,\forall j \in \{1, \dots, n_{i-1}\}\}, \quad C = A \setminus B.$$

Note that $S$ is used in the definition of $B$, not $S'$. The function $W \mapsto y(a, W)$ is differentiable at all $W^* \in B$. This holds because the definition of $B$ and the fact that $B \subset A$ imply that $y(a, W)$ is constant in an open neighbourhood of $W^*$. So $W \mapsto y(a, W)$ is smooth on $B$.

We will now show that $C$ has measure zero. Clearly,

$$C = \bigcup_{i=1}^{H+1} \bigcup_{j=1}^{n_i} C_{i,j},$$

where
$$C_{i,j} = \{W \in A \mid S'_{i,j}(a, W) = 0, \text{ and } S'_{k,l}(a, W) \neq 0 \,\forall k < i, l \in \{1, \ldots, n_k\}$$
$$\exists l \in \{1, \ldots, n_{i-1}\}, \text{ such that } S_{i-1,l,l}(a, W) \neq 0\}$$
We will show that each of the $C_{i,j}$ is contained in the finite union of sets which have Lebesgue measure zero, and this will in turn show that $C$ has measure zero by sub-additivity of measure.

If $W^* \in C_{i,j}$, then $W^*$ is in the zero set of the function
$$y_{i,j}(a, W) = e_j^T W_{i-1}^T S_{i-1}(a, W^*) W_{i-2}^T S_{i-2}(a, W^*) \ldots S_1(a, W^*) W_0^T a. \tag{2}$$
This is a polynomial in $W$, and is non-zero by definition of $C_{i,j}$. Non-zero real analytic functions have zero sets with measure zero [Mit15], so the zero set of this particular polynomial has measure zero. Moreover, as we vary $W^* \in C_{i,j}$ in (2) we get finitely many distinct polynomials, since the switches $S_{i-1}, \ldots, S_1$ take on finitely many values. This proves that $C_{i,j}$ is in a finite union of measure zero sets, and hence $C$ has Lebesgue measure zero. The map $W \mapsto y(a, W)$ is smooth everywhere else, so we are done. $\square$

**Proof of Lemma 2**. Let $\ell$ be smooth in an open neighbourhood $U$ of a point $W^* \in \mathbb{R}^m$. Defining
$$e(a_i, W) = y(a_i, W) - f(a_i),$$
we compute
$$\frac{\partial^2}{\partial w_{kl}^{j2}} \ell(W) = \frac{1}{N} \sum_{i=1}^N \left( \frac{\partial}{\partial w_{kl}^j} y(a_i, W) \right)^2 + e(a_i, W) \frac{\partial^2}{\partial w_{k,l}^{j2}} y(a_i, W),$$
$$= \frac{1}{N} \sum_{i=1}^N \left( \frac{\partial}{\partial w_{kl}^j} y(a_i, W) \right)^2, \tag{3}$$
where (3) follows since each term in $y(a_i, W)$ is locally a polynomial in the weights where each variable has maximum degree 1. The second derivative of $\ell$ with respect to any variable is therefore non-negative, and so
$$\Delta \ell(W) \geq 0 \tag{4}$$
Hence, $\ell$ is a subharmonic function on $U$, and therefore, by the maximum principle for subharmonic functions [McO03], $\ell$ cannot obtain a maximum at $W^*$, unless it is constant on $U$. $\square$

*Remark*: It is easy to see that the proof of Lemma 2 can be generalized to the case of a loss function
$$\tilde{\ell}(W) = \frac{1}{N} \sum_{i=1}^N g(f(a_i), y(a_i, W)),$$
provided $\frac{\partial^2}{\partial y^2} g(f(a_i), y) \geq 0$ for all $i$.

**Proof of Lemma 3**. Let $X \in \mathbb{R}^m$ be a perturbation direction, with $X = (X_0, \ldots, X_H)$, and set $\tilde{y}(a, W + tX)$ as
$$\tilde{y}(a, W + tX) = S_{H+1}(a, W)(W_H + tX_H)^T S_H(a, W) \ldots S_1(a, W)(W_0 + tX_0)^T a,$$
so that
$$\phi(W + tX) = \frac{1}{2N} \sum_{i=1}^N (f(a_i) - \tilde{y}(a_i, W + tX))^2. \tag{5}$$
Let $e(a_i, W + tX) = \tilde{y}(a_i, W + tX) - f(a_i)$; we compute
$$\frac{d^2}{dt^2}|_{t=0} \phi(W + tX) = \frac{1}{N} \frac{d}{dt}|_{t=0} \sum_{i=1}^N e(a_i, W + tX) \frac{d}{dt} \tilde{y}(a_i, W + tX),$$
$$= \frac{1}{N} \sum_{i=1}^N \left( \frac{d}{dt}|_{t=0} \tilde{y}(a_i, W + tX) \right)^2 + e(a_i, W) \frac{d^2}{dt^2}|_{t=0} \tilde{y}(a_i, W),$$
$$\geq \frac{1}{N} \sum_{i=1}^N e(a_i, W) \frac{d^2}{dt^2}|_{t=0} \tilde{y}(a_i, W).$$

Let $\tilde{W}_i(a, W) = S_{i+1}(a, W)W_i S_i(a, W)$ for $i = 0, \ldots, H$, with $S_0 = I_{n_0}$. It is clear that

$$\tilde{y}(a, W + tX) = (\tilde{W}_H + t\tilde{X}_H)^T \ldots (\tilde{W}_0 + t\tilde{X}_0)^T a, \tag{6}$$

where $\tilde{X}_i = S_{i+1}(a, W)X_i S_i(a, W)$. We may proceed to compute derivatives

$$\frac{d}{dt}\tilde{y}(a, W + tX) = \sum_{i=0}^{H}(\tilde{W}_H + t\tilde{X}_H)^T \ldots \tilde{X}_i^T \ldots (\tilde{W}_0 + t\tilde{X}_0)^T a, \tag{7}$$

$$\frac{d^2}{dt^2}|_{t=0}\tilde{y}(a, W + tX) = \sum_{i=0}^{H}\sum_{j \neq i} \tilde{W}_H^T \ldots \tilde{X}_i^T \ldots \tilde{X}_j^T \ldots \tilde{W}_0^T a. \tag{8}$$

Using the triangle inequality, as well as the sub-multiplicative property of matrix norms, we may estimate

$$\left|\frac{d^2}{dt^2}|_{t=0}\tilde{y}(a, W + tX)\right| \leq \sum_{i=0}^{H}\sum_{j \neq i} \|\tilde{W}_H\|_2 \ldots \|\tilde{X}_i\|_F \ldots \|\tilde{X}_j\|_F \ldots \|\tilde{W}_0\|_2 \|a\|_2. \tag{9}$$

Here we have used the fact that the Frobenius norm dominates the matrix norm induced by the Euclidean 2-norm. Neglecting zeroed out columns and rows, we have

$$\left|\frac{d^2}{dt^2}|_{t=0}\tilde{y}(a, W + tX)\right| \leq \sum_{i=0}^{H}\sum_{j \neq i} \|W_H\|_2 \ldots \|X_i\|_F \ldots \|X_j\|_F \ldots \|W_0\|_2 \|a\|_2,$$

$$\leq \sum_{i=0}^{H}\sum_{j \neq i} \|W\|_*^{H-1}\|X\|^2 \|a\|_2,$$

$$= H(H + 1)\|W\|_*^{H-1}\|X\|^2 \|a\|_2.$$

With $\|a_i\|_2 \leq r$ for all $1 \leq i \leq N$, we therefore obtain

$$\frac{d^2}{dt^2}|_{t=0}\phi(W + tX) \geq -\frac{1}{N}\sum_{i=1}^{N}|e(a_i, W)|H(H + 1)\|W\|_*^{H-1}\|X\|^2 \|a_i\|_2,$$

$$\geq -H(H + 1)\|W\|_*^{H-1}\|X\|^2 r \left(\frac{1}{N}\sum_{i=1}^{N}|e(a_i, W)|\right),$$

$$\geq -\sqrt{2}H(H + 1)\|W\|_*^{H-1}\|X\|^2 r \ell(W)^{1/2}.$$

In the last line we used the Cauchy-Schwarz inequality. Recalling $\ell(W) = \phi(W)$, the lemma is proved. $\square$

**Proof of Lemma 4**: Let $W_0$ be a global minimizer of $\ell_\lambda$; a global minimizer must exist because $\ell_\lambda$ is coercive and continuous. We have $\ell_\lambda(W_0) = \epsilon$, and as such

$$\ell(W_0) \leq \epsilon, \frac{\lambda}{2}\|W_0\|^2 \leq \epsilon. \tag{10}$$

Since $\|W\|_* \leq \|W\|$ for all $W$, we have

$$\ell(W_0)^{1/2}\|W_0\|_*^{H-1} \leq \sqrt{\epsilon}\sqrt{2}^{H-1}\frac{\sqrt{\epsilon}^{H-1}}{\sqrt{\lambda}^{H-1}} \tag{11}$$

$$= \sqrt{2}^{H-1}\frac{\sqrt{\epsilon}^{H}}{\sqrt{\lambda}^{H-1}}, \tag{12}$$

$$< \sqrt{2}^{H-1}\frac{\sqrt{\lambda}^{H+1}}{\sqrt{\lambda}^{H-1}}\frac{1}{\sqrt{2}^{H}H(H + 1)r}, \tag{13}$$

$$= \frac{\lambda}{\sqrt{2}H(H + 1)r}. \tag{14}$$

Since the inequality in (13) is strict, there exists $\theta > 0$ such that

$$\ell(W_0)^{1/2}\|W_0\|_*^{H-1} < \frac{\lambda - \theta}{\sqrt{2}H(H+1)r}, \tag{15}$$

and so $W_0 \in U(\lambda, \theta)$. Moreover, since the slack in (13) is independent of $W_0$, the same $\theta$ must work for all global minimizers. Thus $U(\lambda, \theta)$ contains all global minimizers. $\qquad\square$

**Proof of Theorem 3**. To define the sets $B_i$, enumerate the possible configurations of the switches $S_j(a_k, W)$ as $j = 1, \ldots, H+1$ and $k = 1, \ldots, N$ as we vary $W$; for the $i$th configuration set $B_i$ as the closure of all points in $\mathbb{R}^m$ giving those values for the switches. There are finitely many $B_i$'s, and their union is all of $\mathbb{R}^m$, so (MP-11) is clear.

Define $\phi_i : \mathbb{R}^m \to \mathbb{R}$ as equal to $\ell_\lambda$, but with switches held constant, as prescribed by the definition of $B_i$. Each $\phi_i$ is therefore smooth, and (MP-13) holds by definition of $B_i$. By Lemma 3 and the definition of $U(\lambda, \theta)$, we have, for each point $W \in B_i \cap U(\lambda, \theta)$,

$$\frac{d^2}{dt^2}|_{t=0}\phi_i(W + tX) > -(\lambda - \theta)\|X\|^2 + \lambda\|X\|^2 = \theta\|X\|^2,$$

which proves (MP-12) for all $W \in B_i \cap U(\lambda, \theta)$. The definition of $U(\lambda, \theta)$ uses strict inequalities, so there exists open $V_i \supset B_i \cap U(\lambda, \theta)$ such that each inequality holds for $\phi_i$ on $V_i$, and therefore (MP-12) holds in $V_i$. $\qquad\square$

**Proof of Lemma 5**: We will abbreviate $U(\lambda, \theta)$ as $U$. Let $W \in U$ be a differentiable critical point of $\ell_\lambda$. Suppose that $W$ is not an isolated local minimum, so that there exists $\{W_n\}_{n=1}^\infty \subset U$ all distinct from $W$ such that

$$W_n \to W, \quad \ell_\lambda(W_n) \le \ell_\lambda(W) \quad n = 1, 2, \ldots \tag{16}$$

Let $I \subset \{1, \ldots, L\}$ such that $i \in I$ if and only if $W \in B_i$; $I$ is non-empty by (MP-11). Let $\epsilon > 0$ be small enough that

$$B(W, \epsilon) \subset \bigcap_{i \in I^c} B_i^c \cap U \tag{17}$$

where $B(W, \epsilon)$ is the Euclidean ball of radius $\epsilon$ centred at $W$; such an $\epsilon$ exists because the $B_i$ are closed, and therefore their complements are open. Equation (17) implies

$$B(W, \epsilon) \subset \bigcup_{i \in I}(B_i \cap U), \tag{18}$$

and therefore $\ell_\lambda$ is always equal to one of the $\phi_i$ for $i \in I$ on $B(W, \epsilon)$. Note also that

$$W \in \bigcap_{i \in I} B_i \cap U \subset \bigcap_{i \in I} V_i, \tag{19}$$

and therefore, decreasing $\epsilon$ if necessary, we also have

$$B(W, \epsilon) \subset \bigcap_{i \in I} V_i. \tag{20}$$

This is possible because the $V_i$ are open. We conclude that $\phi_i$ is strongly convex on $B(W, \epsilon)$ for all $i \in I$. Now, let $n$ be large enough that $W_n \in B(W, \epsilon)$. Take $\gamma : [0, 1] \to U$ as

$$\gamma(t) = (1 - t)W_n + tW, \tag{21}$$

so that $\gamma(0) = W_n$, and $\gamma(1) = W$. By assumption,

$$\ell_\lambda(\gamma(0)) \le \ell_\lambda(W). \tag{22}$$

Define

$$t^* = \sup\{t \in [0, 1] \mid \ell_\lambda(\gamma(s)) \le \ell_\lambda(W) \quad \forall s \in [0, t]\}. \tag{23}$$

It is clear that $t^* \ge 0$, by (22), and we claim that $t^* = 1$. Proceeding by contradiction, suppose $t^* < 1$. Then we must have that $\ell_\lambda(\gamma(t^*)) = \ell_\lambda(W)$, and there exists a sequence $\delta_n > 0$ converging to 0 such that

$$\ell_\lambda(\gamma(t^* + \delta_n)) > \ell_\lambda(W) \quad n = 1, 2, \ldots. \tag{24}$$

Let $J \subset \{1, \ldots, L\}$ such that $i \in J$ if and only if $\gamma(t^*) \in B_i$. Note that $J \subset I$ by (17). Again, since the $B_i$ are closed, there exists $\delta > 0$ such that for

$$\gamma(t) \in \left( \bigcup_{i \in J^c} B_i \right)^c, \quad \forall \quad t \in [t^*, t^* + \delta). \tag{25}$$

This implies $\ell_\lambda(\gamma(t)) \in \{\phi_i(\gamma(t)) \mid i \in J\}$ for all $t \in [t^*, t^* + \delta)$. Note, however, that

$$\phi_i(\gamma(t)) < \ell_\lambda(W), \quad \forall \quad t \in (t^*, t^* + \delta), i \in J. \tag{26}$$

This holds because $\phi_i(\gamma(t^*)) = \phi_i(W) = \ell_\lambda(W)$ for all $i \in J$, and $\phi_i$ is a strongly convex function on $B(W, \epsilon)$. As such, $\ell_\lambda(\gamma(t)) < \ell_\lambda(W)$ for all $t \in (t^*, t^* + \delta)$, contradicting (24). We conclude that $t^* = 1$, and so $\ell_\lambda(\gamma(t)) \leq \ell_\lambda(W)$ for all $t \in [0, 1]$. Let $\{t_k\}_{k=1}^\infty \subset [0, 1]$ be a sequence converging to 1 such that for all $k$, $\gamma(t_k) \in B_i$ for some $i \in I$; such an $i$ must exist because there are finitely many $B_i$ and infinitely many points in $[0, 1]$. Because $\phi_i$ is strongly convex with parameter $\theta$,

$$\phi_i(\gamma(t_k)) \geq \phi_i(W) + \langle \nabla \phi_i(W), \gamma(t_k) - W \rangle + \frac{\theta}{2} \|\gamma(t_k) - W\|^2. \tag{27}$$

Since $\phi_i(\gamma(t_k)) - \phi_i(W) \leq 0$, $\frac{\theta}{2}\|\gamma(t_k) - W\|^2 > 0$, and $\gamma(t_k) - W = (1 - t_k)(W_n - W)$, we obtain

$$0 > \langle \nabla \phi_i(W), W_n - W \rangle. \tag{28}$$

We also have

$$\frac{\phi_i(\gamma(t_k)) - \phi_i(W)}{\|\gamma(t_k) - W\|} = \frac{\ell_\lambda(\gamma(t_k)) - \ell_\lambda(W)}{\|\gamma(t_k) - W\|} \tag{29}$$

As $t_k \to 1$, the right hand side converges to 0 since $W$ is a differentiable critical point of $\ell_\lambda$. On the other hand,

$$\lim_{t_k \to 1} \frac{\phi_i(\gamma(t_k)) - \phi_i(W)}{\|\gamma(t_k) - W\|} = \frac{\langle \nabla \phi_i, (W_n - W) \rangle}{\|W_n - W\|} < 0, \tag{30}$$

which is a contradiction. We therefore conclude that if $W$ is a differentiable critical point, it is an isolated local minimum. $\square$

**Proof of Lemma 6**: Assume by contradiction that $W$ is a local minimum, but that there exists a sequence of points $\{W_n\}_{n=1}^\infty$ satisfying

$$W_n \to W, \quad \ell_\lambda(W_n) = \ell_\lambda(W) \quad n = 1, 2, \ldots \tag{31}$$

In the proof of Lemma 5, we have shown that if (31) holds, then for all $n$ large enough, there are points on the segment joining $W_n$ and $W$ obtaining strictly smaller values of $\ell_\lambda$; this is shown explicitly in (26). This is shown without assuming that $\ell_\lambda$ is differentiable at $W$, and thus we may use it here. We therefore obtain a sequence of points $\tilde{W}_n$ on the segments connecting $W_n$ to $W$, satisfying

$$\ell_\lambda(\tilde{W}_n) < \ell_\lambda(W), \tag{32}$$

and therefore $W$ cannot be a local minimum, since $\tilde{W}_n$ converges to $W$. This contradiction proves the result. $\square$

**Proof of Lemma 7**: For linear networks, all previous results hold with $L$, the number of sets $B_i$, equal to 1. We therefore conclude that Lemma 5 holds for linear neural networks. Suppose by contradiction that $\ell_\lambda$ has a critical point $W \neq 0$ in $U(\lambda)$. Then there exists $\theta$ such that $W \in U(\lambda, \theta)$. By Lemma 5, $W$ is an isolated local minimum. Since $W \neq 0$, there exists $i \in \{0, \ldots, H\}$ such that $W_i \neq 0$. Let $R : \mathbb{R}^{n_{i+1}} \to \mathbb{R}^{n_{i+1}}$ be a rotation. Consider the weight

$$\tilde{W} = (W_0, W_1, \ldots, W_i R^T, R W_{i+1}, \ldots, W_H). \tag{33}$$

Since $R^T = R^{-1}$, it is clear that

$$\ell_\lambda(\tilde{W}) = \ell_\lambda(W), \tag{34}$$

and there are rotations $R$ such that $\tilde{W} \neq W$ since $W_i \neq 0$. Taking $R$ a small rotation, we can make $\tilde{W}$ arbitrarily close to $W$, and therefore $W$ is not an isolated local minimum. This contradiction proves the result. $\square$

*Remark*: The same proof may not work in the case of a non-linear network, as the switches may interfere with the rotation matrix $R$.

**Proof of Lemma 8**: We have

$$\frac{d}{dt}\gamma(t) = \langle \nabla \ell_\lambda(W(t)), \dot{W}(t) \rangle = -\|\nabla \ell_\lambda(W(t))\|^2. \tag{35}$$

Set $u(t) = -\frac{d}{dt}\gamma(t)$; by assumption, $u(t)$ is $C^1$ on $[0, t^*]$ and satisfies

$$u'(t) = -\frac{d^2}{dt^2}\gamma(t) = -2\nabla \ell_\lambda(W(t))^T \mathbf{H}(\ell_\lambda(W(t)))\nabla \ell_\lambda(W(t)),$$
$$\leq -C\|\nabla \ell_\lambda(W(t))\|^2,$$
$$= -Cu(t).$$

As such, $u(t)$ satisfies the differential inequality

$$u'(t) \leq -Cu(t), \tag{36}$$

for $t \in [0, t^*]$. This is the hypothesis of Grönwall's inequality, which in this setting has a short proof which we will reproduce for completeness. Let $v(t)$ be the solution to

$$v'(t) = -Cv(t), \quad v(0) = u(0). \tag{37}$$

Assume $u(0) > 0$, since otherwise the conclusion of the lemma is immediate. Then $v(t) = u(0)e^{-Ct} > 0$ for all $t$, We have,

$$\frac{d}{dt}\frac{u(t)}{v(t)} = \frac{u'(t)v(t) - u(t)v'(t)}{v^2(t)},$$
$$= \frac{v(t)(u'(t) + Cu(t))}{v^2(t)} \leq 0.$$

So, $u(t)/v(t)$ is a decreasing function which starts at 1 when $t = 0$. We therefore conclude that for all $t \in [0, t^*]$,

$$u(t) \leq v(t) \Rightarrow \|\nabla \ell_\lambda(W(t))\|^2 \leq \|\nabla \ell_\lambda(W(0))\|^2 e^{-Ct}, \tag{38}$$

which proves the lemma. $\qquad\square$