[Reviews · NeurIPS 2019]

Reviewer 1



Originality: I am not convinced that the contributions of this paper are more significant than that of [1], which have been cited in this paper already. Specifically, in comparison with [1] in Line 82, the authors state that these conclusions apply to a smaller set in weight space. I would appreciate it if the authors could quantify the difference here and have a discussion section to show the comparison with some form of mathematical comparison.              Further, there have been quite a few papers that show convergence of GD on neural networks using something like strong convexity. eg. [3] and I would appreciate a discussion on how that it relates to this. Clarity The paper is written quite clearly and it is easy enough to follow the paper. I specially appreciate the proof sketch given below the statement of the main theorems. But in certain places in the appendix, there is some confusion regarding the scope of the index variables (I have pointed it out in the next section).  However, certain things needs a bit more elaboration eg. Line 214 - “The bound in (6) is restrictive”. Could you elaborate on what the implications of this is?  A minor comment would be using the \|W\|_F notation to represent the Frobenius norm instead of the \| W \| notation, which is commonly used to denote the matrix 2-norm. I am aware that it you have clearly defined the notation though. Technical Comments 1. Line 88-89:], I would request the authors to elaborate on Line 88-89 about “then differentiable local minima should satisfy ….  as the network have enough capacity”. However, [2] deals with a very different setting- including leaky-ReLUs and perhaps more importantly a multiplicative noisy regularizor like dropout and a mild over-parameterization requirement. The maths to connect this work with [2] doesn’t seem entirely straightforward to me. Moreover, the set U here depends on both the spectral norm and the loss.
 2. My first major doubt is in the derivation of (7). If I understand correctly,  you impose the constraint $ \|W\|_* \le R$ into (6) to obtain (7). However, $ \|W\|_* \le R =>  1/ (\|W\|_*) \ge 1/R $ and it doesn’t imply (7). In that case, I think your claim in Line 176 that U(\lambda, \theta) \cap B(R) doesn’t include all points with training error below that certain bound.
 3. Could you please clarify, what you mean by “bounded region” in Line 196 - “ ..  in bounded region, as prescribed by the set U(\lambda, \theta) “ ? Which bounded region are you referring to ?
 4. In Line 198, that the size of U(\lambda,\theta) depends on the training data. Could you clarify, if you refer to whether a small enough loss can be obtained by a small enough W (in terms of spectral norm) ? It would be very useful in somehow quantifying/giving an idea of how large this set U(\lambda, \theta) actually is. 
 5. Lemma 5,6 : Lemma 5 and 6 says that every critical point is an isolated local minima. However, it doesn’t guarantee the prescece of one such local minima does it ? Particularly, because the loss is regularized it is not trivial to expect a zero-loss and thus I don’t see how you can guarantee that U(\lambda,\theta) is simply not a trivial set which doesn’t contain any minimizers ?
 6. Could you please elaborate on  Line 214 - “The bound in (6) is restrictive”? Do you mean that the set defined by U in (6) is practically uninformative ?
 7. Further the experiments in (4) are quite far from providing evidence for the theory:
 1. They do not talk about strong convexity but only when restricted to the gradient path. These are quite different.
 2. They work with exponential loss which do not have a finite minimizer instead of quadratic losses, which do have finite minimizer. In some sense, the exponential loss are not expected to have any minima in any bounded set of parameters. So, it can never satisfy the hypothesis of minima in a bounded set.
 8. In Supplementary Line 11 - 15, the indices i,j are quite confusing. They seem to be referring to different things. Please make it clearer.
 9. I don’t understand how you derived (3) from the previous line in the suppmentary. Could you please elaborate why it is okay to ignore the second term ?
 10. Line 56 in Supplementary: Definition of U(\lambda, \theta) also admits a strict inequality. So, 15 should also have a strict inequality.
 11. In the proof of Lemma 5 in Line 69, you start by assuming that W is a critical point. It doesn’t guarantee the existence of one. Could you please clarify if this is not true ?
 In summary, I find the claim of the paper to not quite match the results of the paper in that it is a overstatement of the result for reasons pointed above. For this reason, I am rejecting the paper but I will be willing to change my score if you can argue against my points above.   [1] Safran, Itay, and Ohad Shamir. "On the quality of the initial basin in overspecified neural networks." International Conference on Machine Learning. 2016. [2] Soudry, Daniel, and Yair Carmon. "No bad local minima: Data independent training error guarantees for multilayer neural networks." arXiv preprint arXiv:1605.08361 (2016). [3] Du, Simon S., et al. "Gradient descent finds global minima of deep neural networks." arXiv preprint arXiv:1811.03804 (2018). ---------------------------------------------------------- Thank you for answering my questions in the rebuttal and for pointing out my mistake in understanding the derivation of (7). Based on the fact that it was my misunderstanding and not a problem with the paper that had initially caused me to give lower scores and upon discussion with other reviewers, I am now updating my score to reflect this. I hope you will incorporate the rest of changes and corrections we talked about. I do like the contributions of the paper and I hope the authors would continue working in this field and potentially identify cases where the theory merges better with empirical findings, which in my opinion is the strongest drawback of the paper. Other than that, I would like the author to discuss the applicability of Lemma 4 as to, if it ever kicks in and identify problems where the $\lambda$ actually needs to be large and the pre-requisities of the lemma kick in.

Reviewer 2



This paper is well-written and makes novel contributions to the analysis of the energy landscape of neural networks (although, admittedly, I am not an expert on this topic). The author's results contribute to the understand of the optimization problems to be solved during training and have the potential to foster new convergence results for gradient descent on such problems. The experimental section is, however, weaker than the theoretical one. It is a little unfortunate that the necessary bound (6) was too difficult to satisfy in practise even on small toy problems such that the numerical experiments cannot confirm the theory. This clearly limits the significance of the findings for practical applications. Nevertheless, the experiments could lead to the conjecture that for different problems (classification with cross-entropy, softmax, batchnorm, etc.) similar properties hold. The criteria investigated in the experiments, namely convexity along a trajectory, is weaker than convexity of the (high dimensional) energy landscape, and works that have computed the Hessian after the convergence of gradient descent often still report a few negative eigenvalues (e.g. https://arxiv.org/abs/1611.07476). Thus, I am wondering to what extend theory and numerical experiments actually demonstrate the same effect. I this respect it seems unnatural to even deviate from the setting of a regression problem to a classification problem. In any case, I think it would be interesting to consider a network with few enough parameters to actually compute the Hessian and its eigenvalues. Despite the weak link between the experiments and the theory, I consider the theory to clearly be the main contribution, which is why I see this submission on the positive side.

Reviewer 3



I kind of like this paper. This is the first theoretical work, at least in my sight, that does not apply any unrealistic assumption and toy architecture of the deep network. The model studied here is both deep and nonlinear, which is something that the society is actually using. If there is no other unrealistic assumption hiding at somewhere I have not noticed, I would give strong credit. The piecewise strong convexity is a nice and reasonable notation to characterize the network, and something that aligns with my personal conjecture. Too bad, I am not the one to propose and prove it. The paper is written well and math seems to be solid and clean. I wonder whether the theory could work to explain some existing empirical findings. E.g. what is the relation of learning rate with the number of layers, and what is the relation with the number of nodes per layer, without the overparameterization assumption. This would make a huge difference if the theory is applied to some nontrivial findings.

[Author Response · NeurIPS 2019]

**Response to Reviewer 1**: Thank you for your detailed review. In turn, we will provide detailed clarifications in the hope that these will persuade you to revisit your verdict and increase your score for our paper.

First, our results are not subsumed by [1] (we use your reference numbering). Their most relevant results show that 2-layer networks have a loss function which, on some convex sets, resembles a convex function in that all local minima are global minima, and all sublevel sets are connected. These two properties do not imply convexity, however; $f(x, y) = x^2 y^2$ satisfies these conditions and is nowhere convex. Our results, on the other hand, imply local strong convexity on certain sets, which is a stronger conclusion. They also hold for any network architecture, as opposed to the 2-layer networks of [1]; this is a unique contribution of our paper, since many theoretical results in this area use restricted architectures. Conversely, the results of [1] hold on all of weight space, while our results hold on $U(\lambda, \theta)$, a proper subset which is nonetheless important as it can contain all global minimizers of the problem (see Lemma 4).

The connection between our work and [3] is tenuous; the Gram matrix they study only uses first derivatives of the predictions with respect to the weights and so says nothing about the convexity of the loss function.

Our responses to your technical comments use your enumeration. We have not responded to points we do not dispute.

1. The results of [2] hold for almost all noise realizations, and after selecting a particular realization, small modifications of our results hold. In this setting, only our calculation of the second derivatives of the loss needs to be modified, but roughly the same proof works with a leaky ReLU provided the slope $s$ for negative inputs is bounded.

2. You've misunderstood our derivation of (7). Define $A(R, \lambda, \theta)$ as the set on the right hand side of (7). Then

$$W \in A(R, \lambda, \theta) \quad \text{implies} \quad ||W||_* \leq R \quad \text{and} \quad \ell(W)^{1/2} < \frac{\lambda - \theta}{\sqrt{2}H(H+1)rR^{H-1}}$$

From these two conditions we have $A(R, \lambda, \theta) \subset B(R)$ and

$$\ell(W)^{1/2}||W||_*^{H-1} \leq \ell(W)^{1/2}R^{H-1} < \frac{(\lambda - \theta)R^{H-1}}{\sqrt{2}H(H+1)rR^{H-1}} = \frac{(\lambda - \theta)}{\sqrt{2}H(H+1)r} \Rightarrow A(R, \lambda, \theta) \subset U(\lambda, \theta).$$

3. Here we are referring to the bounded region $A(R, \lambda, \theta)$. We will make this clearer in the paper.

4. If the training data and network architecture is such that zero non-regularized error is possible, then a small enough loss can be obtained at some $W$ regardless of the size of $||W||_*$; see the definition of $U(\lambda, \theta)$ to understand why.

5. Lemmas 5 and 6 do not guarantee a local minimum of $\ell_\lambda$ in $U(\lambda, \theta)$. Lemma 4 does, under some conditions.

6. The training process typically halts before the bound is satisfied. Thus, our work may describe the region around a global minimizer which gradient descent does not typically reach. But, see our response to Reviewer 2 for the details of a new experiment we performed in response to their comments which shows the relevance of this bound.

7a. These are distinct but related concepts. Strong convexity is sufficient for strong convexity restricted to gradient paths, but not necessary. Some useful properties of convexity still hold if a function is strongly convex restricted to gradient paths. For example, some guarantees of convergence rate for gradient descent still hold.

7b. Note that we use weight decay, so the loss becomes coercive, and a finite global minimizer is guaranteed.

9. As we mention, $y(a_i, W)$ is locally a polynomial in the weights where the maximum degree of each variable is 1, like $f(x, y, z) = xy + xyz$; differentiating this twice with respect to the same variable always gives 0.

11. You are correct that this doesn't guarantee the existence of a critical point, but this is not necessary for the validity of the lemma. The existence of a critical point could be guaranteed, for example, by Lemma 4.

**Response to Reviewer 2**: Thank you for your thoughtful review. Based on your comments we have conducted a simple regression experiment with target function $f(x) = 0$ and 100 data points sampled uniformly in $[-1, 1]$. We used an architecture of $H = 1, n_1 = 2$, no biases, no ReLU on the output, and weight decay parameter $\lambda = 1$. Despite having a trivial target function, this problem is non-convex. Experiments show that in 100 independent trials, the bound given in (6) is satisfied for some $\theta$ for $51.6\% \pm 24.0\%$ (mean $\pm$ standard deviation) of the loss change over training - see the paper for the definition of this metric. Hence, gradient descent does enter the set $U(\lambda, \theta)$, but only after some descent.

This experiment provides a stronger connection between the theory and numerics. It is considerably simpler than the simplest example we tested before, which had $H \geq 2$. We will include the details in the final version of the paper.

**Response to Reviewer 3**: We are grateful for your kind words about our paper. We studied the deep and non-linear networks used in practice with no unrealistic assumptions, and we are thankful that you appreciate this.

In terms of using our results to explain existing empirical findings, there is one comment we can make. Empirically, we observed that the addition of batch normalization helped the gradient descent trajectory enter the piecewise strongly convex regime much sooner than without it (compare the results for MNIST to those for CIFAR10 and 100). A possible conjecture is that batch normalization convexifies the loss function; this is an interesting avenue for future research.

[Meta-Review · NeurIPS 2019]

This paper shows that the quadratic loss with weight decay of deep ReLU networks is piecewise strongly convex on a nonempty open set where every critical point is a local minimum, and every local minimum is isolated. Initially the paper received mixed reviews, with two positive and one negative review. On the positive side, the contribution is found to be quite significant because it analyzes realistic networks (deep and non-linear). On the other hand, one reviewer had issues with the proof, and another with the experiments. The rebuttal addressed the issues raised by the reviewers, and the negative review updated the score. Specifically, the reviewer acknowledged he/she had miss-understood the proof. Upon discussion, the reviewers agreed that the paper should be accepted.